# The induction of preterm labor in rhesus macaques is determined by the strength of immune response to intrauterine infection

Monica Cappelletti[1☯], Pietro Presicce[1☯], Ma Feiyang[2,3], Paranthaman Senthamaraikannan[4], Lisa A. Miller[5,6], Matteo Pellegrini[2,3], Myung S. Sim[7], Alan H. Jobe[4], Senad Divanovic[8], Sing Sing Way[9], Claire A. Chougnet[8], Suhas G. Kallapur[1]*

1 Divisions of Neonatology and Developmental Biology, UCLA Mattel Children's Hospital, David Geffen School of Medicine, University of California, Los Angeles, Los Angeles, California, United States of America, 2 Department of Molecular, Cell and Developmental Biology Medicine, University of California, Los Angeles, Los Angeles, California, United States of America, 3 Institute for Quantitative and Computational Biosciences–Collaboratory, University of California, Los Angeles, Los Angeles, California, United States of America, 4 Division of Neonatology and Pulmonary Biology, University of Cincinnati College of Medicine, Cincinnati, Ohio, United States of America, 5 California National Primate Research Center, University of California Davis, Davis, California, United States of America, 6 Department of Anatomy, Physiology, and Cell Biology, School of Veterinary Medicine, University of California Davis, Davis, California, United States of America, 7 Department of Medicine Statistics Core, University of California, Los Angeles, Los Angeles, California, United States of America, 8 Division of Immunobiology, Cincinnati Children's Hospital Research Foundation, University of Cincinnati College of Medicine, Cincinnati, Ohio, United States of America, 9 Infectious Diseases, Cincinnati Children's Hospital Research Foundation, University of Cincinnati College of Medicine, Cincinnati, Ohio, United States of America

☯ These authors contributed equally to this work.
* skallapur@mednet.ucla.edu

**Data Availability Statement:** File S1_Data.xlsx contains numerical values used to generate the graphs in the figures. RNAseq data are available in

## Abstract

Intrauterine infection/inflammation (IUI) is a major contributor to preterm labor (PTL). However, IUI does not invariably cause PTL. We hypothesized that quantitative and qualitative differences in immune response exist in subjects with or without PTL. To define the triggers for PTL, we developed rhesus macaque models of IUI driven by lipopolysaccharide (LPS) or live *Escherichia coli*. PTL did not occur in LPS challenged rhesus macaques, while *E. coli*–infected animals frequently delivered preterm. Although LPS and live *E. coli* both caused immune cell infiltration, *E. coli*–infected animals showed higher levels of inflammatory mediators, particularly interleukin 6 (IL-6) and prostaglandins, in the chorioamnion-decidua and amniotic fluid (AF). Neutrophil infiltration in the chorio-decidua was a common feature to both LPS and *E. coli*. However, neutrophilic infiltration and *IL6* and *PTGS2* expression in the amnion was specifically induced by live *E. coli*. RNA sequencing (RNA-seq) analysis of fetal membranes revealed that specific pathways involved in augmentation of inflammation including type I interferon (IFN) response, chemotaxis, sumoylation, and iron homeostasis were up-regulated in the *E. coli* group compared to the LPS group. Our data suggest that the intensity of the host immune response to IUI may determine susceptibility to PTL.

a public repository (GEO Accession: GSE181054) https://www.ncbi.nlm.nih.gov/geo/query/acc.cgi?acc=GSE181054.

**Funding:** This study was supported by U01 ES029234 (CAC), Burroughs Wellcome grant (CAC), CCHMC Perinatal Infection and Inflammation Collaborative (CAC), R21HD90856 (SGK), and R01HD 98389 (SGK), VWR foundation (MC). The funders had no role in study design, data collection and analysis, decision to publish, or preparation of the manuscript.

**Competing interests:** The authors have declared that no competing interests exist.

**Abbreviations:** Abx, antibiotics; AF, amniotic fluid; CFU, colony-forming unit; DEG, differentially expressed gene; DSCF, Dwass–Steel–Critchlow–Fligner; FDR, false discovery rate; HIER, heat-induced antigen retrieval; IA, intra-amniotic; IFN, interferon; IgG, immunoglobulin G; IHC, immunohistochemistry; IL-1, interleukin 1; IL-1β, interleukin 1 beta; IL-6, interleukin 6; IL-8, interleukin 8; IM, intramuscular; IRF, interferon regulatory factor; IUI, intrauterine infection/inflammation; LAL, Limulus Amebocyte Lysate Assay; LPS, lipopolysaccharide; mAb, monoclonal antibody; MPO, myeloperoxidase; NF-kB, nuclear factor kappa B; NK, natural killer; NKT, natural killer T; PCA, principal component analysis; PRR, pattern recognition receptor; PTL, preterm labor; qPCR, quantitative polymerase chain reaction; RIN, RNA integrity number; RNA-seq, RNA sequencing; SUMO, small ubiquitin–like modifier; TLR4, Toll-like receptor 4; TNFα, tumor necrosis factor alpha.

## Introduction

Intrauterine infection/inflammation (IUI) most commonly originates in the lower genitourinary tract and ascends to the uterine cavity including the chorio-decidual space and the amniotic cavity. About 40% of preterm labor (PTL) cases are associated with IUI [1]. Importantly, the causal link between IUI and PTL is well established [2]. In a prospective PTL biomarker study, only 25% of women with increased amniotic fluid (AF) interleukin 6 (IL-6) had detection of microorganisms in the AF [3]. Thus, both intrauterine infection and inflammation can cause PTL. Currently, the rate of prematurity remains stubbornly high at about 10% of all United States of America births [4], causing 75% of perinatal mortality and 50% of the long-term morbidity [1]. Apart from maternal morbidity, IUI causes fetal inflammation and increases the risk for fetal and newborn brain, gut, and lung injury in both clinical studies and in animal models (reviewed in [5]). Although IUI is a frequent cause of PTL, about 25% to 45% of pregnancies at risk for PTL (including cases with presumed IUI) do not delivery preterm within 14 days of presentation [3,6–8]. Thus, PTL is not an invariable consequence of IUI.

What factors during IUI lead to crossing the threshold of PTL are not known. Human studies of PTL have implicated pro-inflammatory agents such as interleukin 1 (IL-1), IL-6, interleukin 8 (IL-8) and tumor necrosis factor alpha (TNFα), prostaglandins, metalloproteinases, and the alarmin HMGB1 as potential etiologic factors [2,3,9,10]. However, large unbiased proteomic or genome-wide association studies have not yielded clinical useful biomarkers of PTL [11,12]. A caveat with transcriptomic studies in both humans and animal models of PTL is that the controls often have no or very little inflammation [13,14]. Pathways leading to IUI tend to confound the analyses of PTL.

In experimental models different doses, route of administration, preparations of lipopolysaccharide (LPS), and microorganisms can lead to different responses to both intrauterine inflammation and PTL (reviewed in [15]). At low doses, intrauterine LPS may cause intrauterine and fetal inflammation but not PTL [16]. We recently reported that lower doses of LPS given by intra-amniotic (IA) route compared to intraperitoneal route trigger PTL in mice [17]. Similarly, higher doses of ultrapure LPS are needed compared to the less refined preparations (used in this study) to trigger PTL in mice [17], presumably because the less pure preparations engage more receptors than Toll-like receptor 4 (TLR4). Microbial invasion of AF with group B streptococci is a more potent inducer of PTL compared to localized chorio-decidual infection in nonhuman primates [18–20]. In our previous dose–response studies in sheep, we demonstrated that there is a plateau phase beyond the 10-mg IA LPS for intrauterine inflammation [21,22]. Sheep maternal and fetal weights at comparable gestation and AF volumes are higher by a factor of 7 to 10 compared with rhesus macaques. Previous studies in a chronically instrumented model of rhesus macaque reported intrauterine inflammation and PTL after 0.01 to 0.1 mg *Escherichia coli* sourced LPS by IA route [23]. We used a higher 1-mg IA *E. coli* LPS dose to simulate more plateau type responses that would not be sensitive to small changes in LPS exposure. We previously reported that this 1-mg IA injection of *E. coli* LPS or interleukin 1 beta (IL-1β) in pregnant noninstrumented rhesus macaques caused IUI but not PTL [24–26], with characteristics that phenocopied human chorioamnionitis [27]. *E. coli* is the leading cause of maternal sepsis during pregnancy [28], yet experimental models evaluating how *E. coli* prenatal infection impacts parturition timing remain limited. For example, while *E. coli* infection during pregnancy was recently shown to efficiently cause fetal wastage and congenital fetal invasion in mice, the effects on PTL were not reported, likely given the relatively short pregnancy duration in rodents compared with larger mammalian species [29]. We now report that IA injection of live *E. coli* causes IUI and PTL. Importantly, live *E. coli*–induced PTL was

**Table 1. PTL frequency after IUI in rhesus.**

| Agonist | Dose | Duration of exposure | Maternal weight (kg) | PTL |
|---|---|---|---|---|
| Saline (control) | 1 ml | 48 hours | 8.9 ± 1.5 | 0/21 |
| LPS (from *E. coli*) | 1 mg | 48 hours | 8.4 ± 1.1 | 0/8 |
| LPS (from *E. coli*) | 1 mg | 5 days[#] | 7.48 ± 0.93 | 0/9 |
| Live *E. coli* | $1 \times 10^6$ CFU | 48 hours | 8.8 ± 2.1 | 5/5[**] |

[**] $p < 0.01$ versus IA LPS (Fisher exact test).

[#] This group of animals was used only to evaluate whether an extended IA LPS exposure (i.e., 5 days) results in PTB.

CFU, colony-forming unit; IA, intra-amniotic; IUI, intrauterine infection/inflammation; LPS, lipopolysaccharide; PTB, preterm birth; PTL, preterm labor.

not rescued by antibiotics (Abx), recapitulating the lack of efficacy of prenatal Abx in reducing PTL in humans [30,31].

Thus, our 2 models of LPS-induced IUI and live *E. coli*–induced PTL with IUI offer unique opportunities to unravel the IUI specific pathways responsible for induction of PTL. We tested the hypothesis that quantitative and qualitative thresholds in immune response exist in subjects with or without PTL.

## Results

### New rhesus model of IUI with PTL: IA *E. coli* injection

*E. coli* was used for the studies since this organism is a prototypic invasive perinatal pathogen and a major cause of early neonatal sepsis resulting from maternal IUI [32]. In our new model of IUI, pregnant rhesus macaque were inoculated IA with live *E. coli* ($1 \times 10^6$ colony-forming unit [CFU]) at 80% to 85% of term gestation (term is approximately 165 days gestational age). PTL was monitored for 48 hours after LPS or live *E. coli* after which surgical delivery was performed (S1 Fig). In contrast to IA LPS causing no PTL within 48 hours, infection with IA *E. coli* caused PTL in 5/5 dams within 48 hours of injection (Table 1). To confirm lack of PTL in the IA LPS group, we extended the period of observation to 5 days in another set and confirmed no PTL in 0 out of 9 animals (Table 1). Furthermore, IA *E. coli* caused maternal bacteremia in 3/5 dams and fetal bacteremia in 100% of pregnancies (Table 2).

### LPS and *E. coli* induce similar immune infiltration in the chorioamnion-decidua

To characterize IUI, we analyzed immune cell infiltration in the chorioamnion-decidua in different rhesus models of IUI. Myeloperoxidase (MPO+) neutrophils were not detected in the controls but readily detected in the chorio-decidua after LPS or *E. coli* exposure (Fig 1A). Additionally, *E. coli* induced neutrophil infiltration in the amnion (Fig 1A). In chorio-decidua, flow cytometry demonstrated that neutrophils and macrophages were the most abundant immune cells after LPS and *E. coli* exposures, in terms of both absolute counts (Fig 1B) and frequency (S2 Fig; gating strategy is shown in S3 Fig). Neutrophil frequency and counts were

**Table 2. *E. coli* invasion in tissue compartments.**

| Agonist | AF culture + (at delivery) | CD culture + | Maternal blood culture + | Fetal blood or liver culture + |
|---|---|---|---|---|
| Live *E. coli* | 3/3 | NA | 3/5 | 5/5 |

AF, amniotic fluid; CD, chorio-decidua; NA, not analyzed.

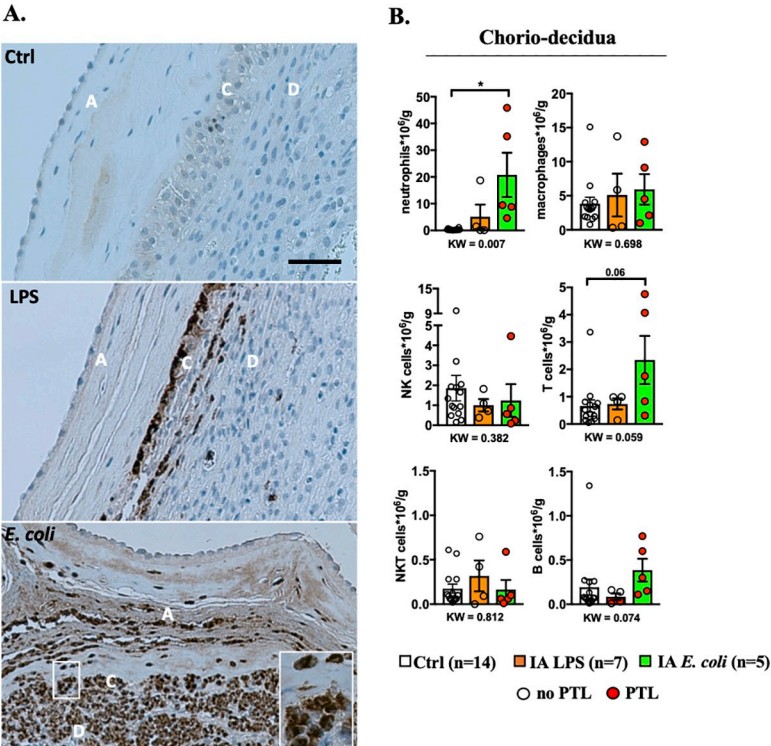

**Fig 1. Increased neutrophil infiltration in chorio-decidua upon IA LPS or *E. coli* exposure. (A)** One representative MPO staining of 4/group showing neutrophil infiltration in fetal membranes after LPS or live *E. coli* exposure but not in controls (amnion A, chorion C, and decidua D). Note that the LPS induced accumulation of neutrophils at the chorio-decidua junction but not in the amnion. IA *E. coli* injection induced neutrophil infiltration in the amnion in addition to the chorio-decidua (scale bar: 50 μm). **(B)** Chorio-decidua cell suspensions were analyzed by multiparameter flow cytometry and the different leukocyte populations were defined as indicated in S3 Fig. IA *E. coli* exposure increased significantly the count of chorio-decidua neutrophils that represent the major immune cell population. A slight but not significant increase in count of T cells with no changes for macrophages, NK cells and B cells compared to the control animals is observed. Cell count is expressed per gram of tissue. Red circles denote animals with PTL, while clear circles denote animals without PTL. Data are mean, SEM. A *p*-value for 3 group comparison by KW nonparametric test is shown in each panel, with post hoc analysis by DSCF method showing *$p < 0.05$ between comparators. See S1 Data for numerical values. DSCF, Dwass–Steel–Critchlow–Fligner; IA, intra-amniotic; KW, Kruskal–Wallis; LPS, lipopolysaccharide; MPO, myeloperoxidase; NK, natural killer; PTL, preterm labor.

higher after *E. coli* compared to LPS exposure (Fig 1B, S2 Fig). Macrophage and natural killer (NK) cell frequency decreased after either LPS or *E. coli* exposure, but absolute counts were similar to controls (Fig 1B, S2 Fig). T-cell absolute count was higher with borderline significance (Fig 1B) in the IA *E. coli* group compared to saline controls (*p* = 0.06) (Fig 1B), but their frequency was similar among the treatments (S2 Fig). Natural killer T (NKT) cells frequency decreased compared to controls and B cell frequency or absolute counts did not significantly change in the LPS and *E. coli* groups (Fig 1B, S2 Fig). Overall, both models induced similar immune infiltration with a higher neutrophil frequency and absolute counts after IA *E. coli* compared to IA LPS exposure.

## *E. coli* induces higher pro-inflammatory cytokines and prostaglandins in the AF compared to LPS

We determined the level of inflammation induced in the different IUI models. In comparison to controls, all of the mediators tested increased after LPS or *E. coli* exposure in the

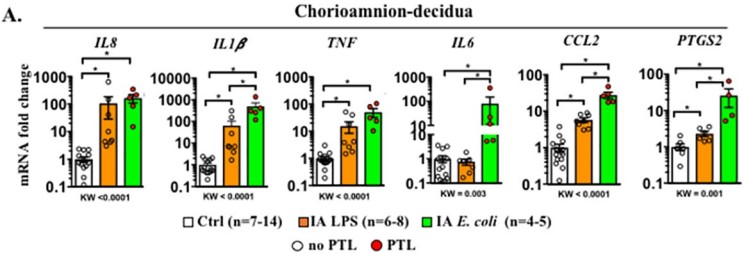

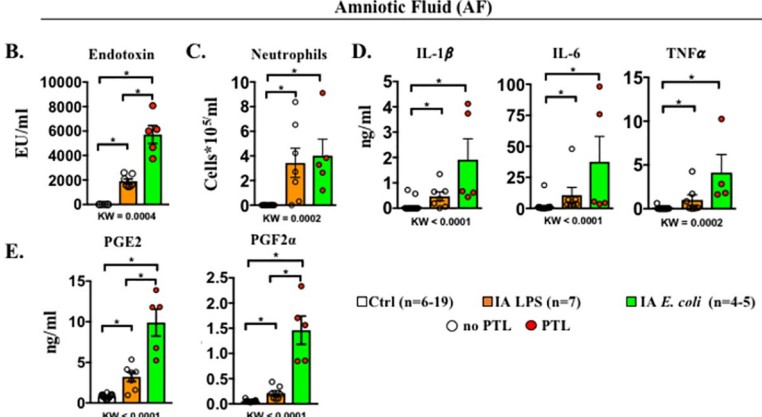

**Fig 2. Higher inflammation in the chorioamnion-decidua and in the AF by live *E. coli* compared to LPS. (A)** Chorioamnion-decidua inflammatory cytokine mRNAs in different rhesus IUI models were measured by qPCR. Average mRNA values are fold increases over the average value for the control group after internally normalizing to the housekeeping 18S RNA. AF levels of **(B)** endotoxin levels measured by limulus lysate assay, **(C)** neutrophil differential counts on cytospins, and **(D)** inflammatory cytokines, and **(E)** prostaglandins. Compared to IA LPS exposure, IA *E. coli* exposure induced a higher expression of *IL1β*, *IL6*, *CCL2*, and *PTGS2* mRNAs in the chorioamnion-decidua, higher AF endotoxin levels, and higher prostaglandin levels in the AF. Data are mean, SEM. A *p*-value for 3 group comparison by KW nonparametric test is shown in each panel, with post hoc analysis by DSCF method showing *$p < 0.05$ between comparators. In the *E. coli* group (green bar), red circles denote animals with PTL, while clear circles denote animals without PTL. See S1 Data for numerical values. AF, amniotic fluid; DSCF, Dwass–Steel– Critchlow–Fligner; IA, intra-amniotic; IL-1β, interleukin 1 beta; IL-6, interleukin 6; IUI, intrauterine infection/ inflammation; KW, Kruskal–Wallis; LPS, lipopolysaccharide; PTL, preterm labor; qPCR, quantitative polymerase chain reaction; TNFα, tumor necrosis factor alpha.

chorioamnion-decidua with the exception of *IL6*, which only increased in the *E. coli* group (Fig 2A). *IL-1*β, *CCL2*, *IL6*, and *PTGS2* expression significantly increased in the chorioamnion-decidua of *E. coli* group compared to LPS (Fig 2A). Endotoxin levels in the AF were significantly higher after IA *E. coli* compared to IA LPS (Fig 2B). Given that neutrophils were the predominant immune cells in the inflamed chorioamnion-decidua, we next examined their levels in the AF. The numbers of AF neutrophils increased comparably in both *E. coli* and LPS groups (Fig 2C). We then compared the cytokine responses in the AF of *E. coli* versus LPS groups. Levels of AF IL-1β, TNFα, and IL-6 were all significantly higher in animals inoculated with *E. coli* and LPS groups compared to controls, with a trend toward higher but not statistically significantly different levels in the *E. coli* compared to LPS group (Fig 2D). Similarly, the prostaglandins PGE2 and PGF2α increased in both groups compared to controls but, we observed 2- to 3-fold higher concentration in the AF from *E. coli* versus LPS animals (Fig 2E). In contrast to large increases of cytokines in the AF, the changes in maternal plasma were much more modest. *E. coli* and LPS exposure increased IL-6 levels in the maternal plasma but did not change IL-8, CCL2, or TNFα levels (S4A Fig). The fetal plasma IL-6 and TNFα

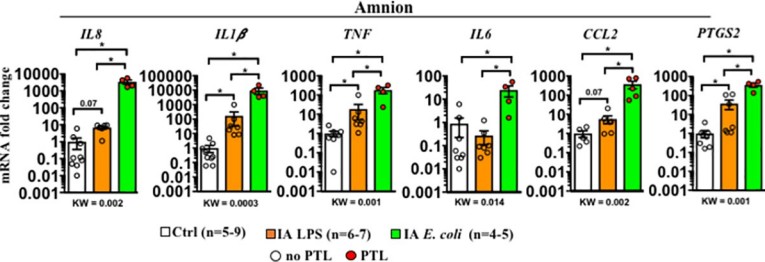

**Fig 3. Higher inflammation in the amnion by live *E. coli* compared to LPS.** Inflammatory cytokine mRNAs in different rhesus IUI models were measured in surgically isolated amnion tissue by qPCR. Average mRNA values are fold increases over the average value for the control group after internally normalizing to the housekeeping 18S RNA. Data are mean, SEM. A *p*-value for 3 group comparison by KW nonparametric test is shown in each panel, with post hoc analysis by DSCF method showing \*p < 0.05 between comparators. Red circles denote animals with PTL, while clear circles denote animals without PTL. See S1 Data for numerical values. DSCF, Dwass–Steel–Critchlow–Fligner; IA, intra-amniotic; IUI, intrauterine infection/inflammation; KW, Kruskal–Wallis; LPS, lipopolysaccharide; PTL, preterm labor; qPCR, quantitative polymerase chain reaction.

increased slightly but significantly after LPS. Although we only had 2 samples available due to nonscheduled delivery, *E. coli* exposure also increased fetal plasma cytokines (S4B Fig).

Differential up-regulation of mediators in the *E. coli* versus LPS groups could be due to time-dependent effects. We previously reported that IA LPS induction of intrauterine inflammation is higher at 16 hours compared to 48 hours [27]. We therefore compared key genes differentially expressed in the *E. coli* group with LPS exposure of 16 hours (using samples archived from our previous study [25,27] and S1 Table). Similar to the results for the IA LPS 48 hours group, *E. coli* induction of chorioamnion-decidua expression of *IL6* and *CCL2* was higher compared to controls, and AF prostaglandin levels were higher compared to IA LPS 16 hours (S5A and S5B Fig). These results suggest that higher induction of key mediators of IUI after IA *E. coli* compared to LPS may not be explained by temporal trajectories of gene expression.

## *E. coli* drives higher levels of inflammation in the amnion

Because *E. coli* but not LPS induced neutrophil infiltration in the amnion (Fig 1A), we analyzed the anatomic locations of inflammatory response. We surgically separated the amnion from the chorio-decidua. In comparison to controls, all of the mediators tested increased after LPS or *E. coli* exposure in the amnion with the exception of *IL6*, which did not increase after LPS exposure (Fig 3). In comparison to the LPS group, *E. coli* increased amnion expression of *IL-1β*, *IL8*, *TNFα*, *CCL2*, *IL6*, and *PTGS2* mRNA levels. Together, these data show that inflammation in the amnion is significantly more pronounced after *E. coli* compared to LPS exposure.

## Abx treatment does not decrease *E. coli*–driven inflammation and preterm birth

To control bacteremia and thus simulate clinical situations, we added Abx treatment in another set of pregnant rhesus infected with *E. coli* (IA *E. coli* + Abx group; S6 Fig). Abx treatment with cefazolin + enrofloxacin starting 24 hours after IA *E. coli* effectively eradicated maternal bacteremia in 7/8 subjects based on negative cultures 1 day after starting Abx and at the end of the experiment. In one subject, there was a persistent amniotic and fetal bacteremia. None of the dams infected with *E. coli* developed significant fever (S7A Fig) or severe abnormality in the peripheral white blood cell count (S7B Fig). Despite the clearance of organisms in

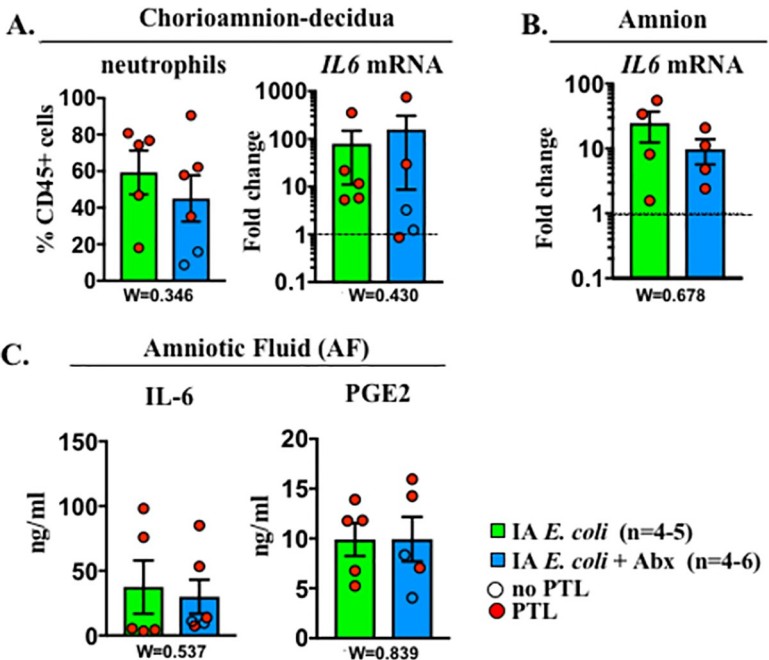

**Fig 4. Abx treatment did not decrease *E. coli*–induced inflammation.** The comparison groups are IA *E. coli* without Abx (green bar) and IA *E. coli* with Abx (blue bar). Red circles denote animals with PTL, while clear circles denote animals without PTL. Values were expressed as fold change value for the IA saline control group. Comparison of **(A)** chorio-amnion decidua neutrophil frequency and IL6 mRNA, **(B)** amnion IL6 mRNA, and **(C)** AF IL6 and PGE2. Dashed line represents the mean value of ctrl animals. Data are mean, SEM. Wilcoxon test (W) comparing the 2 groups was applied, and *p*-values are shown under each panel. See S1 Data for numerical values. Abx, antibiotics; AF, amniotic fluid; IA, intra-amniotic; IL-6, interleukin 6; PGE2, prostaglandin E2; PTL, preterm labor.

the Abx group, PTL was observed in 6/8 subjects by day 3. To understand if antimicrobial therapy also reduced inflammation, we compared select mediators in different compartments between IA *E. coli* and IA *E. coli* + Abx groups. The tested mediators were comparable between the 2 groups: neutrophil frequency and *IL6* expression in the chorioamnion-decidua (Fig 4A), I*L6* expression in the amnion (Fig 4B), and IL-6 and PGE2 levels in the AF (Fig 4C). In sum, the Abx treatment did not materially affect the pregnancy outcome and the extent of inflammation after *E. coli* infection.

## Anatomical localization of *IL6* and *PTGS2* genes in the fetal membranes

Since *IL6* and prostaglandins were the key differentially expressed genes (DEGs) in the chorioamnion-decidua between *E. coli* +/− Abx and LPS groups (Fig 2A), we colocalized *IL6* or *PTGS2* with MPO, a marker of activated neutrophils. We used dual RNAscope fluorescence to visualize the *IL6* or *PTGS2* mRNA expression coupled with immunofluorescence to detect MPO+ cells. We successfully executed this method by first completing the RNA scope study on sections followed by traditional immunohistology on the same tissue section (see Materials and methods section for details). This tandem dual assay allows simultaneous target gene visualization with immunolocalization of protein of interest in situ. Samples from *E. coli* +/− Abx were combined because they did not show any significant differences (Fig 4). MPO+ cells infiltrated the chorio-decidua tissue similarly in both LPS and *E. coli* +/− Abx groups (Fig 5A and 5B), confirming the previous immunohistochemistry (IHC) findings (Fig 1A). However, in the amnion, MPO+ cells were rarely detected in the LPS group but were abundant in the *E. coli* +/− Abx group (Fig 5A–5C), again confirming IHC previous findings (Fig 1A). The majority

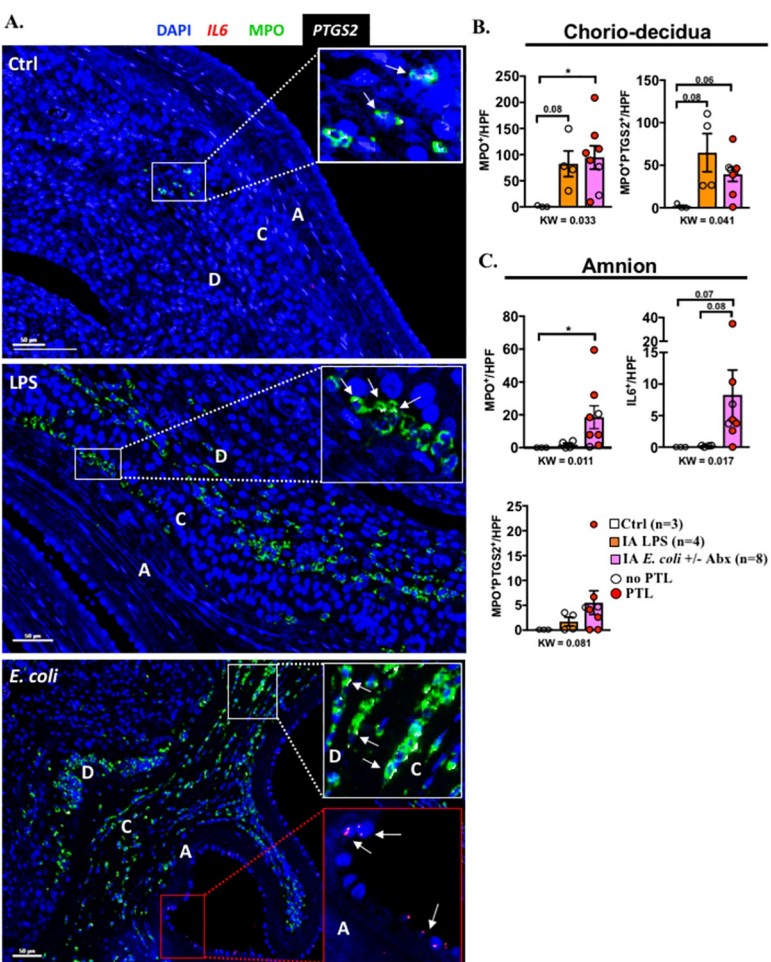

**Fig 5. Cellular localization of *IL6* and *PTGS2* in the fetal membranes. (A)** Representative multiplex fluorescence detection of *IL6* and *PTGS2* mRNA identified by RNAscope in situ hybridization and MPO colocalization by immunofluorescence. *IL6* is shown in red, *PTGS2* in white, and MPO in green. DAPI indicates nuclear staining (blue) in all images. A = amnion, C = chorion, and D = decidua. White arrows in the white insets indicate colocalization of MPO (green) and PTGS2 (white) in the chorio-decidua. White arrows in the red inset indicate *IL6*+ epithelial and mesenchymal cells (red) in the amnion. Quantification of the cells expressing different markers in the **(B)** chorio-decidua and **(C)** amnion. Average of 5 randomly selected HPF fields were plotted as the representative value for the animal. Counts were performed in a blinded manner. Red circles denote animals with PTL, while clear circles denote animals without PTL. Data are mean, SEM. A *p*-value for 3 group comparison by KW nonparametric test is shown in each panel, with post hoc analysis by DSCF method showing *$p < 0.05$ or a borderline *p*-value between comparators. See S1 Data for numerical values. Abx, antibiotics; DSCF, Dwass–Steel–Critchlow–Fligner; HPF, high-power field; IA, intra-amniotic; LPS, lipopolysaccharide; KW, Kruskal–Wallis; MPO, myeloperoxidase; PTL, preterm labor.

(>80% to 90%) of PTGS2+ cells were also MPO+ both in the LPS and *E. coli* groups, suggesting that chorio-decidua neutrophils are a major source of prostaglandin production in the fetal membranes during IUI (Fig 5B). MPO+ cells in the amnion of the *E. coli* exposed animals also expressed PTGS2, with far fewer MPO+PTGS2+ cells in the amnion of the LPS group (Fig 5C). In contrast to the chorio-decidua being a major source of PTGS2 expression, IL6+ cells were exclusively present in the amnion of the *E. coli* +/− Abx group (Fig 5C), consistent with our quantitative polymerase chain reaction (qPCR) data (Fig 3). Notably, MPO+ cells did not express IL6. Rather, IL6+ cells morphologically appeared to be amnion epithelial cells and amnion mesenchymal cells (Fig 5A). Taken together, these observations indicate that neutrophil recruitment and PTGS2 expression in the chorio-decidua is a common feature to both

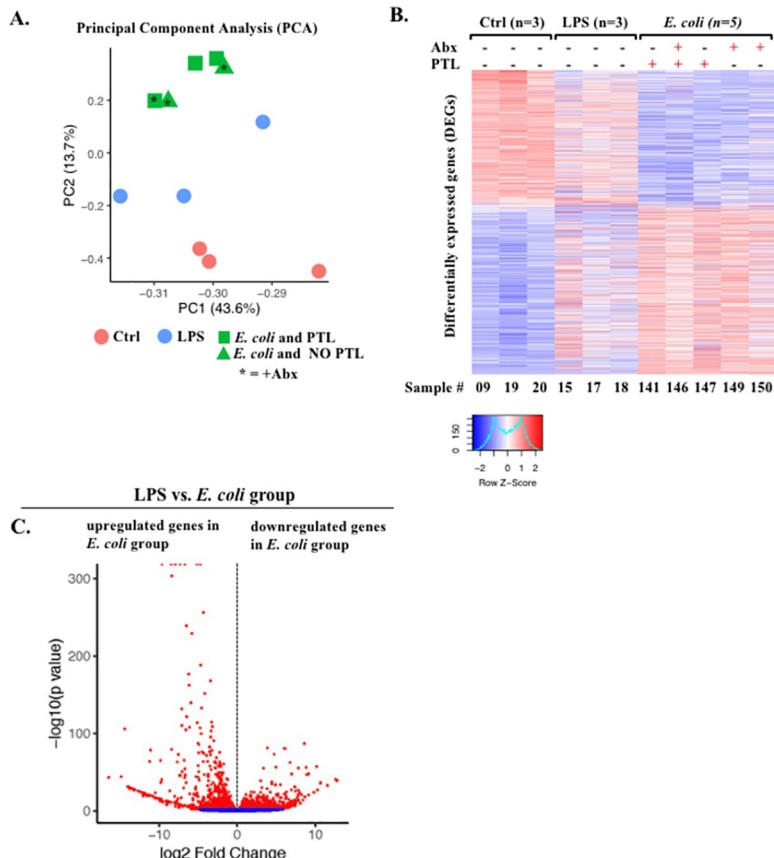

**Fig 6. Comparative transcriptomics in chorioamnion-decidua between IA LPS and *E. coli* exposure. (A)** PCA of RNA-seq data from chorioamnion-decidua tissue showing different segregation of transcriptomic profile based on exposures. **(B)** Heatmap of genes that are differentially expressed in chorioamnion-decidua cells upon different treatments showing the minimal interanimal variability within each group. **(C)** Volcano plot displaying DEGs from *E. coli* animals compared with LPS animals. Red dots indicate genes with FDR adjusted *p*-value <0.05. Abx, antibiotics; DEG, differentially expressed gene; FDR, false discovery rate; IA, intra-amniotic; LPS, lipopolysaccharide; PCA, principal component analysis; PTL, preterm labor; RNA-seq, RNA sequencing.

LPS and *E. coli* +/− Abx immune response. However, neutrophilic PTGS2 expression and IL6 expression in the amnion is differentially induced by *E. coli* infection.

## *E. coli* induces specific pathways in chorioamnion-decidua involved in exacerbation of inflammation

To gain transcriptomic insights associated with PTL, we compared RNA sequencing (RNA-seq) analysis using the chorioamnion-decidua from the different animal groups. For these studies, we combined RNA-seq analyses from *E. coli* (*n* = 3) and *E. coli* + Abx (*n* = 2) since both had similar inflammatory response (Fig 4). Unbiased principal component analysis (PCA) of global gene expression profiles demonstrated distinct segregation of *E. coli* versus LPS groups (Fig 6A). Interestingly, profiles for *E. coli* only and *E. coli* + Abx clustered together (Fig 6A), consistent with our previous findings (Fig 4). Heatmaps of DEGs demonstrated distinct profiles in *E. coli* versus LPS groups (Fig 6B). Importantly, the replicates within each group were similar, demonstrating the consistency of the findings (Fig 6B). Comparison of *E. coli* versus LPS chorioamnion-decidua samples revealed 3,301 DEGs (false discovery rate [FDR] adjusted *p*-value < 0.05) (Fig 6C). A total of 1,415 genes were up-regulated (fold change ≥2) by *E. coli* compared to the LPS (Fig 7A).

Gene ontology category analysis revealed that the pathways up-regulated by *E. coli* (representative genes) include positive regulation of chemotaxis (*CXCL3*), type I interferon (IFN) α/β receptor signaling (*IRF7*), global genome nucleotide excision repair (*SUMO1*), and iron homeostasis (*FHT1*) (Fig 7B). Moreover, 1,102 genes were down-regulated (fold change ≤1/2) by *E. coli* compared to LPS (Fig 7A). The major pathway down-regulated by *E. coli* was posttranslational modifications (e.g., ubiquitination and acetylation) (S8A Fig). The unchanged genes (fold change 1/2 < × < fold change 2) were 784 and were involved in pathways regulating the mRNA processing, splicing, and protein deacetylation (S8B Fig).

## Pathways for PTL induction

Within the *E. coli* exposed group, only 2/8 animals did not undergo PTL (both received Abx). These 2 samples gave us the opportunity to extend our transcriptomic analyses by restricting the comparison between RNA-seq analyses from chorioamnion-decidua of animals infected by *E. coli* with or without PTL. Despite the paucity of samples, our data of PCA of global gene expression profiles clearly demonstrated distinct clustering and heatmaps of PTL versus no PTL cases (S9A and S9B Fig). We found 1,331 up-regulated genes in PTL compared to no PTL samples (FDR adjusted *p*-value of 0.05; fold change ≥2) (S10A Fig). These genes were enriched in biological processes such as inflammatory response, regulation of IL-6, IL-1, and TNF and other cytokine production, neutrophil-mediated immunity and degranulation, nuclear factor kappa B (NF-kB) signaling, and response to type I IFN (S10A Fig). Down-regulated genes (fold change ≤1/2) in the PTL samples were enriched in cellular response (S10B Fig), notch signaling, and T differentiation, while mRNA-related processing and posttranslational modification remained unaltered (fold change 1/2 < × < fold change 2) (S10C Fig).

## Discussion

Viviparous species have evolved a strategy of inflammation inducing labor [33,34]. A corollary of the inflammation hypothesis is that there is a threshold of inflammation that triggers labor

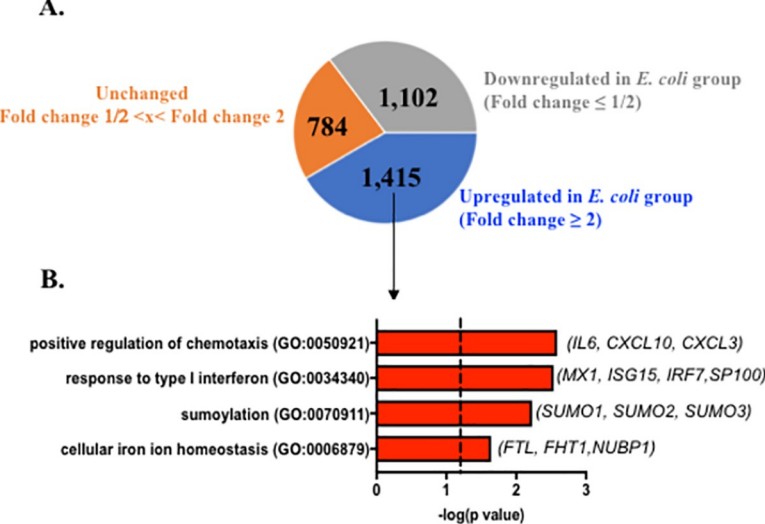

**Fig 7. Up-regulated biological processes (GO terms) in chorioamnion-decidua by live *E. coli*.** **(A)** Pie chart displaying DEGs up-regulated (fold change ≥2), unchanged (fold change 1/2< x < fold change 2), and down-regulated in *E. coli* (fold change ≤1/2) with FDR adjusted *p*-value <0.05 compared to LPS. **(B)** Biological processes significantly up-regulated in chorioamnion-decidua of *E. coli* treated animals compared to LPS exposure. Representative genes of the biological processes are shown in parenthesis. DEG, differentially expressed gene; GO, gene ontology; FDR, false discovery rate; LPS, lipopolysaccharide.

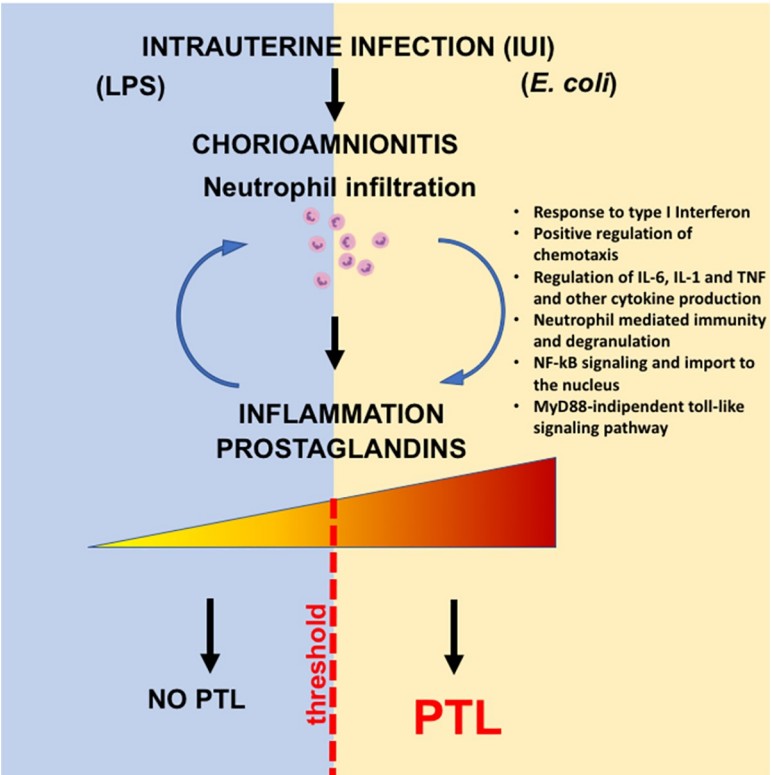

**Fig 8. Working model: IUI is a major risk factor for PTL.** Chorioamnionitis is the hallmark feature of IUI and is typically characterized by the infiltration of neutrophils into the fetal membranes. Evidence suggests that the extent of bacterial colonization, route of infection, and the stimulatory capacity of the bacteria all play key roles in the activation of pro-inflammatory signaling cascades that induce production of pro-inflammatory cytokines (e.g., IL-6) and chemokines (e.g., CCL2). The intensity of immune response is a key player for the outcome of the pregnancy, which, in turn, promote prostaglandin production leading to PTL. IL-1, interleukin 1; IL-6, interleukin 6; IUI, intrauterine infection/inflammation; NF-kB, nuclear factor kappa B; PTL, preterm labor; TNF, tumor necrosis factor.

[14,35]. Whether these concepts extend to inflammation-mediated PTL is not known. The peculiar anatomy of the female genital tract lends itself to vulnerability for ascension of the lower genital organisms in the upper genital space. An exuberant immune response to invading pathogens can incur collateral tissue injury and PTL [36]. Therefore, the host immune system must balance the risk of inflammation induced PTL with protection from infectious organism. Although maternal bacteremia is not very common, *E. coli* infection is a leading cause when it occurs and can have devastating consequences both to the mother and the fetus [28]. In a mouse model, pregnancy was associated with higher susceptibility to *E. coli* infection compared to virgin mice [29]. To gain insights into the pathogenesis of PTL, we used nonhuman primate models of IUI with and without PTL (with or without *E. coli* infection) simulating human chorioamnionitis. We suggest that both the magnitude of inflammation and activation of certain pathways (e.g., IL-6, CCL2, and prostaglandins) play a key role in triggering PTL (Fig 8).

Although a number of pro-inflammatory cytokines are up-regulated both by LPS and live *E. coli*, IL-6 and prostaglandins levels increased significantly more after *E. coli*. Mice with IL6 genetic knockout have delayed increases in prostaglandins, delayed onset of normal parturition despite on time progesterone withdrawal, and resistance to LPS induced preterm delivery [37,38]. In humans, higher AF IL-6 predicted earlier preterm delivery [3,39]. However,

exogenous administration of IL-6 alone did not induce PTL in either mice or rhesus macaques [40,41]. Thus, IL-6 signaling in the context of an inflammatory response seems to be required to trigger PTL. A number of different knockouts for genes in the prostaglandin signaling pathway in mice demonstrates the critical need for prostaglandin signaling in lysis of the corpus luteum of the ovary, leading to progesterone withdrawal and normal parturition in mice [42]. In humans, prostaglandin levels increase prior to parturition and administration of prostaglandins can induce PTL [43,44]. Thus, IL-6 and prostaglandin differentially up-regulated by *E. coli* being potential factors for causing PTL is strongly supported by biologic rationale.

Differential responses to inflammatory stimuli may include different cell/tissues participating and/or different quantitative or qualitative host response. We observed that neutrophil infiltration in the amnion and *IL6* expression in the amnion was present after live *E. coli* stimulation but not after LPS injection in the AF. Histological demonstration of neutrophils in the amnion denotes a higher stage of chorioamnionitis in humans compared to neutrophils only in the chorio-decidua [45]. Although amnion cells can express *PTGS2* during normal labor [46], the source of prostaglandins during inflammation mediated PTL is not well understood. Both in our LPS and live *E. coli* models, we demonstrated that activated neutrophils expressing MPO were the major cells expressing inducible PTGS2 expression in the fetal membranes. Interestingly, *IL6* expression in the amnion was detected after live *E. coli* but not after LPS exposure. The cellular source of *IL6* was amnion epithelial cells and amnion mesenchymal cells with no contribution from the infiltrating neutrophils.

The quality and quantity of signals that trigger different inflammatory response at the maternal fetal interface are not well explored. Innate immune cells are known to modulate their responses based on input stimuli. As an example, the dynamics of NF-kB activation in macrophages depends on which pattern recognition receptor (PRR) signaling is triggered and whether a combination of PRRs are signaled [47]. Consistently, in our study, live *E. coli* with a combination of different PRRs induced a higher expression of IL6 and some type I IFN responsive genes in the fetal membranes compared to LPS alone. Another mechanism that might explain a higher inflammatory response with live organism is that bacterial mRNA is known to induce a more potent innate immune response [48]. Effector cells in the host can mount a tiered inflammatory response by switching on or off specific modules of transcription factors in response to different signals [49]. Important classes of transcription factor differentially regulated by live *E. coli* include regulation of chemotaxis, type I IFN axis, sumoylation, and iron homeostasis (Fig 7).

Using genetic knockout mice and antibody neutralization, we previously demonstrated that type I IFN axis primes LPS responses on maternal hematopoietic cells, increase the expression of *IL6* and *TNF*, and increase susceptibility to preterm birth [38]. Furthermore, we observed that type I IFN priming of LPS responses are conserved in nonhuman primates and humans [38]. Type I IFN signaling can also increase the chemoattractant CXCL10-driven neutrophil recruitment to the site of inflammation [50]. Thus, the higher type I IFN response in live *E. coli* may be an important driver of PTL in our study.

Sumoylation is a posttranslational modification by small ubiquitin–like modifiers (SUMOs) proteins that regulates innate immune response predominantly by negative regulation of interferon regulatory factor (IRF), type I IFN responses, and inflammation [51–53]. Our data suggest that *E. coli* increased the SUMO gene expression in the fetal membranes, which would be expected to decrease inflammation. This is counter to the increased inflammation observed in our study. However, SUMO expression is increased in the placenta in preeclampsia and chorioamnionitis [54,55]. Since sumoylation regulates function of a vast array of proteins, more work is needed to precisely understand how SUMO genes regulate inflammation during PTL.

Inflammation has a potent effect on iron homeostasis. Known as hypoferremia of inflammation, the cytokine-driven increase in hepcidin, largely mediated by IL6, decreases iron transport into plasma [56]. The net effect is to decrease availability of non-transferrin–bound free iron, which stimulates the growth of certain pathogenic bacteria. We recently demonstrated that in a rhesus macaque model of IUI induced by LPS, the fetus responded by rapidly up-regulating hepcidin and lowering iron in fetal blood, without altering AF iron status [57]. The present study demonstrates a differential regulation of genes involved in iron homeostasis after live *E. coli* exposure. Overall, the findings suggest that the effects of iron homeostasis are likely due to bacterial infection from an invasive organism and innate host defense response.

Although reproductive anatomy, endocrinology, and immunology are similar in both rhesus macaques and humans, some limitations of our study should be noted. While both IA LPS and live *E. coli* induced IUI, there are differences in dose of LPS exposure in these models. Our previous studies in the sheep demonstrated that IA LPS has an apparent half-life of about 29 hours [58]. On the contrary, live *E. coli* grows quite robustly in the AF. Thus, the levels of LPS decline over time with IA LPS and increase with live *E. coli* organisms. At the end of the experiment, the endotoxin level in the AF were about 3-fold higher in our infection model compared with IA LPS. Thus, some of the differential effects on gene expression could be due to the different dose exposure to LPS in these 2 models. However, neutrophil counts and many cytokines in the AF were similar, suggesting that comparisons are still valid. Although *E. coli* is a common perinatal pathogen, the most common cause of IUI where microorganisms are identified are the Ureaplasma species [15]. Thus, our findings may not be generalizable to all cases of IUI with PTL.

A relatively large number of genes are differentially regulated (both up and down) in the chorioamnion-decidua of LPS versus live *E. coli* exposed rhesus macaques. However, in both screens (LPS versus live *E. coli* and animals undergoing PTL versus no PTL within the *E. coli* exposed group), only a handful of differentially regulated pathways were represented. In both screens, IL6, NF-kB, type 1 IFN, and neutrophil-mediated immunity genes were represented. In sum, our findings shed light on the role of both qualitative and quantitative changes in the regulation of the thresholds of inflammation. Our findings combined with prior studies incriminate these pathways in the pathogenesis of inflammation mediated PTL.

## Materials and methods

### Animals

Normally cycling, adult female rhesus macaques (*Macaca mulatta*) were time mated. At approximately 130 days of gestation (approximately 80% of term gestation), the pregnant rhesus received either 1-ml saline solution ($n = 21$) or 1-mg LPS (Sigma-Aldrich, St. Louis, Missouri, USA) in 1-ml saline solution ($n = 8$). Another group of pregnant animals received live uropathogenic *E. coli* derived from a cystitis clinical isolate (strain UTI89) ($10^6$ CFU in 1 mL) ($n = 5$) at approximately 140 days of gestation (approximately 85% of term gestation). All injections were given by ultrasound-guided IA injection (S1 Fig). In a separate set of *E. coli*–injected animals ($n = 8$), Abx were given starting 24 hours after bacterial injection. The Abx dosing regimen was intramuscular (IM) cefazolin 25 mg/k twice daily + IM enrofloxacin 5 mg/k twice daily. Additionally, IA injection of cefazolin (10 mg) + IA enrofloxacin (1 mg) were given once daily (S6 Fig). Efficacy of Abx was determined by AF and maternal blood cultures 1 day after the start of Abx and in maternal and fetal blood cultures at the end of the experiment. Body temperatures were monitored by an implanted subcutaneous chip (IPTT-300 implantable programmable temperature transponder, BMDS, Seaford, Delaware, USA). Readings were obtained every 12 hours by telemetry only in *E. coli*–infected animals (S7 Fig).

Dams injected with LPS were surgically delivered 48 hours later, while dams infected with *E. coli* had PTL generally about 2 days after IA *E. coli* injection. The animals given Abx delivered preterm 2 to 3 days after *E. coli* injection (*n* = 6) with the remaining 2 animals delivered surgically at 3 days without PTL. Only 1 animal in the *E. coli* group delivered vaginally, and the remainder 4 animals were surgically delivered in active PTL. After delivery, fetuses were euthanized with pentobarbital, and fetal tissues were collected. Another group of dams with LPS were surgically delivered 16 hours later (*n* = 13). Some controls and all the LPS 16 hours and 48 hours animals were used in previous studies [25,27], while all the *E. coli* infection animals have not been previously reported. The animals used for each experiment and their clinical characteristics are listed in S1 Table.

### IHC

Paraffin-embedded chorioamnion-decidua sections were cut at 4-μm thickness and paraffin removed with xylene and rehydrated through graded ethanol. Endogenous peroxidase activity was blocked with 3% hydrogen peroxide in methanol for 10 minutes. Heat-induced antigen retrieval (HIER) was carried out for all sections in AR6 buffer (AR6001KT, Akoya Biosciences, Marlborough, MA, USA) using a Biocare Decloaker at 95°C for 25 minutes. The slides were then stained with against anti-human MPO (1: 3,000; Agilent Dako, Cat#: A0398, Agilent Technologies, Santa Clara, California, USA), and the signal was detected using the Dakocytomation Envision System Labelled Polymer HRP anti-rabbit (Agilent Technologies K4003, ready to use); all sections were visualized with the diaminobenzidine reaction using a VEC-TASTAIN ABC Peroxidase Elite kit (Vector Laboratories, Burlingame, CA, USA) and counterstained with hematoxylin.

### Chorioamnion-decidua dissection

Extraplacental membranes were dissected away from the placenta as previously described [26,27]. After scraping decidua parietalis cells with the attached chorion, the amnion tissue was peeled away from the chorion with forceps. Chorio-decidua cells were washed and digested with Dispase II (Life Technologies, Grand Island, New York, USA) plus collagenase A (Roche, Indianapolis, Indiana, USA) for 30 minutes followed by DNase I (Roche) treatment for another 30 minutes. Cell suspensions were filtered, and the red blood cells lysed and prepared for flow cytometry. Viability was >90% by trypan blue exclusion test. Tissues of fetal membranes were used for RNA analyses described below.

### Flow cytometry of chorio-decidua cells

Monoclonal antibodies (mAbs) used for multiparameter flow cytometry (LSR Fortessa 2, BD Biosciences, San Diego, California, USA) and gating strategy to identify the different leukocyte subpopulations was done as previously described [27] (S3 Fig). Cells were treated with 20 μg/mL human immunoglobulin G (IgG) to block Fc receptors, stained for surface markers for 30 minutes at 4°C in PBS, washed, and fixed in fixative stabilizing buffer (BD Biosciences). Samples were acquired within 30 minutes after the staining. All antibodies were titrated for optimal detection of positive populations and mean fluorescence intensity. At least 500,000 events were recorded for each sample. Doublets were excluded based on forward scatter properties, and dead cells were excluded using LIVE/DEAD Fixable Aqua Dead Cell Stain (Life Technologies). Unstained and negative biological population were used to determine positive staining for each marker. Data were analyzed using FlowJo version 9.5.2 software (TreeStar, Ashland, Oregon, USA).

## Chorioamnion-decidua tissue cytokine quantitative RT-PCR

Total RNA was extracted from snap-frozen chorioamnion-decidua and amnion biopsies after homogenizing in TRIzol (Invitrogen, Waltham, MA, USA). RNA concentration and quality were measured by NanoDrop spectrophotometer (Thermo Fisher Scientific, Wilmington, Delaware, USA). Reverse transcription of the RNA and quantitative RT-PCR were performed using qScript One-Step RT-qPCR Kit (Quanta BioSciences, Beverly, MA, USA) following the manufacturer's instructions and with Rhesus-specific TaqMan gene expression primers (Life Technologies). Eukaryotic 18S rRNA (Life Technologies) was endogenous control for normalization of the target RNAs.

## RNA-seq and analysis

Total RNA was purified and treated with DNase using RNeasy mini kit following manufacturer's recommendation (Qiagen, Valencia, California, USA). After purification, the concentration of total RNA was measured using NanoDrop ND-1000 spectrophotometer (Thermo Fisher Scientific), and the quality was analyzed by the Bioanalyzer 2100 (Agilent Technologies). Samples with RNA integrity number (RIN) $\geq$9.0 were used for mRNA sequencing. RNA library preparation was performed at the DNA Core Facility at Cincinnati Children's Hospital Medical Center. Single-end read sequencing by Illumina HiSeq2500 Ultra-High-throughput sequencing system (Illumina, San Diego, California, USA) was used at an average depth of 20 million reads per sample. Raw sequences were accepted once they passed the quality filtering parameters used in the Illumina GA Pipeline.

The reads were mapped with STAR 2.5.3a to the *M. mulatta* genome (Mmul 8.0.1). The counts for each gene were obtained using quantMode GeneCounts in the STAR commands, and only counts for the featured genes were reserved. Differential expression analyses were carried out using DESeq2. PCA analysis were performed on the rlog counts from DESeq2 using the R function prcomp. Volcano plots were made for the results from the differential expression analyses. Heatmaps were plotted on the log2 value of the normalized counts. Inference of biological processes were generated using Enrichr [59].

## IL6, PTGS2 mRNAs, and MPO staining

RNAscope (RNAscope Fluorescent Multiplexed Reagent Kit, Advanced Cell Diagnostics, USA) was used as per manufacturer's protocol and adjusted for dual detection of mRNA and protein. RNAscope Multiplex Fluorescent Reagent Kit V2 (323100, RNAscope, Advanced Cell Diagnostics) was used according to manufacturer's protocol to label mRNAs for *IL6* and *PTGS2*. Moreover, 5-μm paraffin-embedded sections of the chorioamnion-decidua tissue were stained with positive controls (300040, Advanced Cell Diagnostics), negative controls (320871, Advanced Cell Diagnostics), and probes for targeting Rhesus PTGS2 (497758-C2, Advanced Cell Diagnostics) and Rhesus IL6 (310378, Advanced Cell Diagnostics). Probes were fluorescently labeled with Opal 570 Reagent (Akoya Biosciences, FP1488001KT) and Opal 620 Reagent (Akoya Biosciences, FP1487001KT) and stained with DAPI (4′,6-diamidino-2-phenylindole) to demarcate the nucleus. Following completion of the RNAscope protocol, sections were immunolabeled using a rabbit anti-MPO polyclonal affinity purified antibody (1: 500; Dako Omnis, A0398) and incubated with Dako anti-rabbit HRP polymer (Agilent Technologies K4003) and then conjugated with Opal 690 Reagent (Akoya Biosciences, FP1497001KT). Then slides were digitalized on a Leica Aperio Versa (Leica Biosystems, Vista, California, USA).

## Cytokines and prostaglandins ELISA

Cytokine/chemokine concentrations in AF, fetal, and maternal plasma were determined by Luminex using nonhuman primate multiplex kits (Millipore, Burlington, MA, USA). Lipids were extracted from the AF using methanol to measure prostaglandins PGE2 (Oxford Biomedical Research, Oxford, Michigan, USA) and PGF2α (Cayman Chemical, Ann Arbor, Michigan, USA).

## Endotoxin assay

Endotoxin level in the AF was determined by Limulus Amebocyte Lysate Assay (LAL; Lonza Gampel, Gampel-Bratsch, Switzerland) according to the test procedure recommended by the manufacturer's instructions.

## Statistics

Summary statistics for continuous variables were expressed as means ± SEM. We tested overall difference among 3 groups (Control, IA LPS, and IA *E. coli*) for continuous outcomes using nonparametric Kruskal–Wallis test, and, when there was an evidence of significant difference, we proceeded to pairwise comparisons using Dwass–Steel–Critchlow–Fligner (DSCF) multiple comparison adjustments. Wilcoxon signed rank test and Mann–Whitney test were used to test 2 population comparisons as indicated. Fisher exact test for categorical variables were used to determine differences between groups. Results were considered statistically significant for 2-sided $p < 0.05$. Statistical Analysis Software SAS 9.4 (Cary, North Carolina, USA) was used to analyze the data.

## Study approval and ethics statement

All animal procedures were approved by the Institutional Animal Care and Use Committee (IACUC; protocol # 22121) at the University of California Davis and endorsed by the University of California, Los Angeles. The committee that reviewed and approved is the UC Davis IACUC.

Care and housing of animals met all IACUC, US Department of Agriculture, and US NIH guidelines for humane macaque husbandry, including the presence of enrichment objects, daily foraging enrichment, and auditory and olfactory access to conspecifics in the same room.

## Supporting information

**S1 Fig. A schematic overview of the rhesus model of IUI.** Pregnant rhesus macaques injected with saline or LPS were surgically delivered 48 hours later, while animals infected with *E. coli* had PTL generally about 48 hours after IA *E. coli* injection. IA, intra-amniotic; IUI, intrauterine infection/inflammation; LPS, lipopolysaccharide; PTL, preterm labor.
(TIFF)

**S2 Fig. Chorio-decidua immune cells upon different IA exposures.** Chorio-decidua cell suspensions were analyzed by multiparameter flow cytometry. Cell count is expressed per gram of tissue. Data are mean, SEM. A *p*-value for KW nonparametric test is shown in each panel, with post hoc analysis DSCF method. *$p < 0.05$ between comparators in the post hoc test. Red circles denote animals with PTL, while clear circles denote animals without PTL. See S1 Data for numerical values. DSCF, Dwass–Steel–Critchlow–Fligner; IA, intra-amniotic; KW, Kruskal–Wallis; PTL, preterm labor.
(TIFF)

**S3 Fig. Flow cytometry phenotype of chorio-decidua cells.** Chorio-decidua parietalis cells were scraped, digested with protease/DNAase, and single cell suspensions were used for multi-parameter flow cytometry analysis. Briefly, live cells were first identified by the absence of LIVE/DEAD stain and forward-/side-scatter expression, excluding cell debris. Leukocytes were gated as CD45$^+$ cells.
(TIFF)

**S4 Fig. Inflammatory cytokines in maternal and fetal plasma. (A)** Maternal plasma and **(B)** fetal cord blood inflammatory cytokine in different rhesus IUI models were measured by multiplex ELISA. Compared to control values, IL6 increased in both LPS and IA *E. coli* group, while CCL2 modestly increased in the maternal plasma of IA *E. coli* group. In the fetal plasma, IL-6 increased in both IA *E. coli* and IA LPS groups compared to saline controls with borderline increase in IL12p40 after IA *E. coli*. Data are mean, SEM. A *p*-value using KW nonparametric test is shown in each panel, with post hoc analysis by DSCF method. $^*p < 0.05$ between comparators in the post hoc test. Red circles denote animals with PTL, while clear circles denote animals without PTL. See S1 Data for numerical values. DSCF, Dwass–Steel–Critchlow–Fligner; IA, intra-amniotic; IL-6, interleukin 6; IUI, intrauterine infection/inflammation; KW, Kruskal–Wallis; LPS, lipopolysaccharide; PTL, preterm labor.
(TIFF)

**S5 Fig. Higher IL-6 and CCL2 mRNA expression and prostaglandins by live *E. coli* compared to LPS 16 hours exposure.** Rhesus were treated as in S1 Fig, but LPS exposure was for 16 hours prior to surgical delivery. **(A)** Chorioamnion-decidua inflammatory cytokine mRNAs in different rhesus IUI models were measured by qPCR. Average mRNA values are fold increases over the average value for control after internally normalizing to the housekeeping 18S RNA. **(B)** Prostaglandin levels in different rhesus IUI models were measured by multiplex ELISA in the AF. Data are mean, SEM. A *p*-value using KW nonparametric test is shown in each panel, with post hoc analysis by DSCF method. $^*p < 0.05$ between comparators in the post hoc test. Red circles denote animals with PTL, while clear circles denote animals without PTL. See S1 Data for numerical values. DSCF, Dwass–Steel–Critchlow–Fligner; IL-6, interleukin 6; IUI, intrauterine infection/inflammation; KW, Kruskal–Wallis; LPS, lipopolysaccharide; PTL, preterm labor; qPCR, quantitative polymerase chain reaction.
(TIFF)

**S6 Fig. A schematic of the rhesus treatment models of *E. coli*–driven IUI and Abx treatment.** The Abx dosing regimen starting 24 hours after IA *E. coli* injection to mimic clinical scenario was IM cefazolin 25 mg/k twice daily + IM enrofloxacin 5 mg/k twice daily. Additionally, IA injection of cefazolin (10mg) + IA enrofloxacin (1mg) were given once daily. The animals delivered preterm 2 to 3 days after *E. coli* injection (*n* = 6) with the remaining 2 animals delivered surgically at 3 days without PTL. See S1 Data for numerical values. Abx, antibiotics; IA, intra-amniotic; IM, intramuscular; IUI, intrauterine infection/inflammation; PTL, preterm labor.
(TIFF)

**S7 Fig. *E. coli* infection did not cause fever or severe abnormality in the peripheral white blood cell count in the dam. (A)** Body temperatures were monitored by an implanted subcutaneous chip and readings obtained every 12 hours by telemetry only in *E. coli*–infected animals. **(B)** Peripheral white blood cell count is shown for *E. coli*–infected animals, LPS-injected animals, and control saline. Data are mean, SEM. A *p*-value for KW nonparametric test is shown in panel B comparing the 5 groups of animals at delivery. Mann–Whitney is shown for comparison between preinjection and delivery in the same set of animals as indicated in panel

B. See S1 Data for numerical values. KW, Kruskal–Wallis; LPS, lipopolysaccharide.
(TIFF)

**S8 Fig. Down-regulated and unchanged GO terms in chorioamnion-decidua by live *E. coli*.**
**(A)** Biological processes significantly down-regulated in chorioamnion-decidua of *E. coli*
treated animals (DEGs ≤ fold change 1/2) or **(B)** unchanged (fold change 1/2 < × < fold
change 2) versus LPS. DEG, differentially expressed gene; GO, gene ontology; LPS, lipopoly-
saccharide.
(TIFF)

**S9 Fig. Comparative transcriptomics in chorioamnion-decidua between IA *E. coli* exposure
with or without PTL. (A)** PCA of RNA-seq data from chorioamnion-decidua cells showing
different clustering based on the presence of PTL. **(B)** Heatmap of genes that are differentially
expressed (DEGs) in chorioamnion-decidua cells of animals with PTL versus no PTL. **(C)** Vol-
cano plot displaying genes detected by RNA-seq in chorioamnion-decidua from *E. coli* treated
animals with PTL compared to no PTL animals. DEG, differentially expressed gene; IA, intra-
amniotic; PCA, principal component analysis; PTL, preterm labor; RNA-seq, RNA sequenc-
ing.
(TIFF)

**S10 Fig. Animals undergoing PTL show up-regulated pathways leading to increased
inflammatory response.** Pie chart displaying DEGs up-regulated (fold change ≥2),
unchanged (fold change 1/2 < × < fold change 2) and down-regulated in PTL group (fold
change ≤1/2). GO pathways and representative genes in **(A)** up-regulated in animals with
PTL, **(B)** down-regulated (DEGs ≤ fold change 1/2), or **(C)** unchanged (fold change 1/2 < ×
< fold change 2) in *E. coli* treated animals with versus without PTL. DEG, differentially
expressed gene; GO, gene ontology; PTL, preterm labor.
(TIFF)

**S1 Table. Animal data and sample sizes.**
(PDF)

**S1 Data. Individual quantitative observations underlying the data summarized in the fig-
ures and results.**
(XLSX)

## Acknowledgments

We thank Sarah Davis, Jennifer Kendrick, Sarah Lockwood, Anne Gibbons, Paul-Michael
Sosa, and Marie Jose-Lemoy research personnel at the CNPRC, UC Davis for help with the
animals.

## Author Contributions

**Conceptualization:** Alan H. Jobe, Senad Divanovic, Sing Sing Way, Claire A. Chougnet,
    Suhas G. Kallapur.

**Data curation:** Monica Cappelletti, Pietro Presicce, Ma Feiyang, Paranthaman
    Senthamaraikannan.

**Formal analysis:** Monica Cappelletti, Pietro Presicce, Ma Feiyang, Myung S. Sim.

**Funding acquisition:** Alan H. Jobe, Senad Divanovic, Sing Sing Way, Claire A. Chougnet,
    Suhas G. Kallapur.

**Investigation:** Lisa A. Miller, Matteo Pellegrini, Alan H. Jobe, Claire A. Chougnet, Suhas G. Kallapur.

**Methodology:** Monica Cappelletti, Pietro Presicce, Paranthaman Senthamaraikannan, Matteo Pellegrini.

**Project administration:** Monica Cappelletti, Pietro Presicce, Suhas G. Kallapur.

**Resources:** Lisa A. Miller, Sing Sing Way, Claire A. Chougnet, Suhas G. Kallapur.

**Software:** Pietro Presicce, Ma Feiyang, Matteo Pellegrini.

**Supervision:** Matteo Pellegrini, Alan H. Jobe, Senad Divanovic, Sing Sing Way, Claire A. Chougnet, Suhas G. Kallapur.

**Visualization:** Monica Cappelletti, Pietro Presicce, Ma Feiyang.

**Writing – original draft:** Monica Cappelletti, Pietro Presicce, Suhas G. Kallapur.

**Writing – review & editing:** Monica Cappelletti, Pietro Presicce, Ma Feiyang, Paranthaman Senthamaraikannan, Lisa A. Miller, Matteo Pellegrini, Myung S. Sim, Alan H. Jobe, Senad Divanovic, Sing Sing Way, Claire A. Chougnet, Suhas G. Kallapur.

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
