## [Editor Report · Decision Letter 0]

22 Jan 2021

Dear Dr. Kallapur, 

Thank you for submitting your manuscript entitled "The intensity of the immune response to LPS and E. coli regulates the induction of preterm labor in Rhesus Macaques" for consideration as a Research Article by PLOS Biology.

Your manuscript has now been evaluated by the PLOS Biology editorial staff, as well as by an academic editor with relevant expertise, and I am writing to let you know that we would like to send your submission out for external peer review.

Please re-submit your manuscript within two working days, i.e. by Jan 24 2021 11:59PM.

Kind regards,

Paula

---

Associate Editor

PLOS Biology

---

## [Decision Letter · Decision Letter 1]

23 Feb 2021

Dear Dr. Kallapur,

Thank you very much for submitting your manuscript "The intensity of the immune response to LPS and E. coli regulates the induction of preterm labor in Rhesus Macaques" for consideration as a Research Article at PLOS Biology. Your manuscript has been evaluated by the PLOS Biology editors, an Academic Editor with relevant expertise, and by independent reviewers.

The reviews of your manuscript are appended below. You will see that the reviewers find the work potentially interesting. However, based on their specific comments and following discussion with the academic editor, I regret that we cannot accept the current version of the manuscript for publication. We remain interested in your study and we would be willing to consider resubmission of a comprehensively revised version that thoroughly addresses all the reviewers' comments. We cannot make any decision about publication until we have seen the revised manuscript and your response to the reviewers' comments. Your revised manuscript would be sent for further evaluation by the reviewers.

Having discussed the reviews with the academic editor, we think that it is important that you clearly describe your experiments and clarify the number of animals that were used. Also, regarding the statistics performed, you could do a statistical transformation of your data and then use a parametric method. Given the unequal numbers in the groups it might be better to use GLM rather than conventional ANOVA, as reviewer #2 suggests, but we strongly suggest you to ask for proper statistics advice. Reviewer #1 thinks that the data presentation is confusing and suggests to add a table or flow chart that presents a clear experimental scheme. This reviewer also says that figure 6 as it stands does not support the hypothesis, asks why was fetal plasma collected from only two fetuses, thinks that you should provide additional assurance that the only difference between the LPS and the E. coli treatment is intensity of the inflammation. Reviewer #2 thinks that you should discuss how the initial dose of E. coli or bacterial proliferation post-inoculation equates to the dose of LPS used in the study and review other data on the effectiveness of bacterial infection and LPS inducing preterm labour in animal models including non-human primates in the intro. This reviewer also has questions about the maternal weight in the different groups, says that the estimation of immune cell infiltration by H&E staining is insufficient, the representation of the data in fig 1B is misleading, and is not convinced of the robustness of the data. Reviewer #2 has also further comments that you will need to address. 

We appreciate that these requests represent a great deal of extra work, and we are willing to relax our standard revision time to allow you six months to revise your manuscript. We expect to receive your revised manuscript within 6 months.

**IMPORTANT - SUBMITTING YOUR REVISION**

*Resubmission Checklist*

*Published Peer Review*

*PLOS Data Policy*

*Blot and Gel Data Policy*

Sincerely,

Paula

---

Associate Editor,

pjaureguionieva@plos.org,

PLOS Biology

REVIEWS:

Reviewer #1: Reproductive immunobiology. Sandra E. Reznik, M.D.

Reviewer #2: Mechanisms responsible for the onset of labour

Reviewer #1: This is a comprehensive study, which, for the most part, tells a logical story. 

However, the way the data are presented is confusing. Based on Table 1 there are 5 animals that were administered E. coli. and they all developed PTL. However, in other parts of the paper, it appears that there were 8 animals injected with E. coli. For example, I don't understand these two sentences in the Methods:

The animals given antibiotics delivered preterm 2-3d after E. coli injection (n=6) with the remaining two animals delivered surgically at 3d without preterm labor. Only 1 animal in the E. coli group delivered vaginally and the remainder [sic] 4 animals were surgically delivered in active preterm labor. 

So, were there 5 or 8 animals treated with E. coli? Did they all develop PTL or not? Table 1 is different from Supplementary Table 1.

I think one table or flow chart that presents a clear experimental scheme would greatly strengthen this manuscript.

Is there a mistake in Figure 6? Didn't all 5 E. coli challenged animals develop PTL? Figure 6, as it stands, does not support the hypothesis. But Figure 6 is different from Supplementary Figure 6.

In Supplemental Figure 3B, why was fetal plasma collected from only two fetuses?

Additional assurance that the only difference between the LPS and the E. coli treatment is intensity of the inflammation is needed. In other words, since most of the E. coli challenged macaques developed bacteremia, although it was treated, can we be sure that these bacteremic animals did not have signs and symptoms of sepsis? In other words, might the bacteremic animals have had systemic changes, such as hypoxia, decreased perfusion, metabolic acidosis, etc., which could explain increased rates of PTL? Please provide data showing the absence of clinical stigmata of sepsis or changes in clinical status in the E. coli vs. LPS treated macaques.

Reviewer #2: Overall: This is a well-written, interesting study which addresses mechanisms underlying inflammatory-mediated preterm labor. The study addresses discrepancies by which bacterial infection but not LPS can induce preterm birth in various animal models. The authors suggest that this can be explained by the degree of inflammatory response generated by these agents, and that the greater inflammatory response generated by bacterial is required to induce preterm labor. The manuscript requires revision to address the points raised below. 

Two major issues should be addressed. Firstly, the use of Mann-Whitney U-test for 3 groups is not appropriate and the applicants should consider other tests such as ANOVA. This is important as it may reduce the statistical significance of the data measures. Secondly, the study seeks to compare the effectiveness of bacterial inoculation with LPS administration. However, the authors do not discuss the how the initial dose of E coli (1*106 CFU) or bacterial proliferation post inoculation equates to the dose LPS (1 mg) used in the study. Murine studies show that different doses of LPS and different LPS preparations injected into pregnant dams at late gestation cause utterly different effects. 

Introduction

The introduction is quite short and should review other data on the effectiveness of bacterial infection and LPS inducing preterm labour in animal models including non-human primates.

Results

For instance, how the dose of E coli ( 1*106 CFU) correspond to LPS ( 1 mg)? Mouse studies show that different doses of LPS and different LPS preparations injected into pregnant dams at late gestation cause utterly different effects.

Maternal weight: I assume "in grams" please add

Table 1: Is there any significant difference in Maternal weight at 48h and 5 days (LPS group) or with the E coli group? Any comments??

Line 100, Table 2: AF culture - why only three animals were analyzed? ('3/3")

What happened to CD culture? not detected and not analyzed?

Figure 1A: Estimation of immune cell infiltration by H&E staining is insufficient. Please provide a specific IHC evaluation for neutrophil infiltration as it is done in Figure 5 (i.e. MPO or other available neutrophil markers)

Figure 1B: The figure is slightly misleading - the absolute count of macrophages, NK, T-cells and B cells does not change after E coli vs LPS. ONLY the neutrophils are dramatically increased, which is a logical response to acute infection at 48 hours. I would rather present an absolute count (or counts per unit weight as in supplementary material), not % of CD45.

Line 150, Figure 2B: This "trend " is due to 1 or 2 samples (from 5); the other 3-4 are not different from the control, I'm not convinced on the robustness of these data. Indeed, in the majority of figures the current statistical analysis shows significance only between E. coli and control groups.

Line 172-175 - This is not a sentence and should be rewritten for clarity.

Line 178: P values are shown between TWO groups, while there are THREE groups studied. One-way ANOVA should be used, not T(Mann-Whitney U)-test. Same comment for figure 3

Lines 200-209: Text describes 8 subjects treated with Abs, 6 delivered preterm. Why are only 4-5 are shown on Figures A,B,C?

Lines 220-221: "… IL6 and PGs were the key differentially expressed genes…" - Could the authors clarify what dataset they are referring to for this statement, as it stands this is not clear.

Line 222: Very interesting data - please describe this methodology - I could description of the RNAscope® fluorescence in situ hybridization in the manuscript.

Line 262, Figure 6A : - What analyses were conducted to define clustering? While E.coli animals seem closer together has the limits of the clusters been defined. 

Line 297 : again should consider use of One-way ANOVA

Discussion

Line 336: How do the author explain a significant difference in the neutrophil infiltration without changes in the IL8 expression, and while there is no change in the monocytes, CCL2 is upregulated? 

Lines 370-372 : these data (re: NFkB) are not shown in this manuscript - Please add the data or correct the Discussion. 

Line 379 missing 1 in "Type 1"

Lines 405-412: I think this paragraph could be strengthened. For example "potentially provide possible explanations" and "study provides further insights" are not very informative

The study Strengths and Limitations part is missing.

Methods, 

I couldn't find details of the animals treated with LPS other than that they were previously reported in another study. It would help to provide some details here so that the reader can assess interpret the presented data.

lines 422-426: The authors state that treatment with antibiotics reduced bacteremia but it is not clear how this was confirmed. 

Lines 422, 426-429 - it is hard to understand the number of animals used: Suppl Table 1 shows 7 (not 8) animals, E coli group n=6, but in the text - 4+1.

Line 446: Trypan blue is not the best method to determine viability when using cells for FACS analyses.

Line 523: The authors discuss conducting H&E staining on human fetal membranes. I can't find mention of this in the results and not sure what relevance of such analysis in the current study.

Line 529-534, Mann-Whitney U test for comparisons with multiple groups is not appropriate, the authors should consider using ANOVA.

Line 537: It is usual to provide more details regarding the REB #, etc

---

## [Decision Letter · Decision Letter 2]

21 Jul 2021

Dear Dr. Kallapur,

Thank you for submitting your revised Research Article entitled "The intensity of the immune response to LPS and E. coli regulates the induction of preterm labor in Rhesus Macaques" for publication in PLOS Biology. I have now obtained advice from the original reviewers and have discussed their comments with the Academic Editor. 

Based on the reviews, we will probably accept this manuscript for publication, provided you satisfactorily address the remaining points raised by the reviewers. Please also make sure to address the following data and other policy-related requests.

Reviewer #1 would like you to clarify whether the third group in Table 1 (LPS/ 5 day exposure) is part of the current study.

We suggests a change of the title to better reflect your findings: "Induction of preterm labor in Rhesus macaques is determined by the strength of immune response to intrauterine infection". This is a suggestion, so please modify further if you consider it necessary.

ETHICS STATEMENT:

-- Please include the full name of the IACUC/ethics committee that reviewed and approved the animal care and use protocol/permit/project license. **Please also include an approval number.**

-- Please include the specific national or international regulations/guidelines to which your animal care and use protocol adhered. Please note that institutional or accreditation organization guidelines (such as AAALAC) do not meet this requirement.

DATA POLICY:

Regardless of the method selected, please ensure that you provide the individual numerical values that underlie the summary data displayed in the following figure panels as they are essential for readers to assess your analysis and to reproduce it: Figures 1B, 2ABCDE, 3, 4ABC, 5BC, 6AC and 7B.

**Please also ensure that figure legends in your manuscript include information on where the underlying data can be found, and ensure your supplemental data file/s has a legend.**

**Please ensure that your Data Statement in the submission system accurately describes where your data can be found.**

Please add size bars to the microscopy pictures in figure 1A.

We expect to receive your revised manuscript within two weeks.

*Published Peer Review History*

*Early Version*

Sincerely,

Paula

---

Associate Editor,

pjaureguionieva@plos.org,

PLOS Biology

Reviewer remarks:

Reviewer #1: Sandra E. Reznik, M.D., Ph.D. Reproductive immunobiology.

Reviewer #2: Mechanisms responsible for the onset of labour.

Reviewer #1: The revised version of the manuscript addresses essentially all my previous concerns.

Regarding the basic protocol and animal groups, the revisions and corrections have resolved most of the inconsistencies previously present. I have one remaining small issue there. I don't see where the third group in Table 1 (LPS/ 5 day exposure) is mentioned in either the text or Supplementary Figure 1. Is this group part of the current study?

Reviewer #2: The authors have substantially revised the manuscript to address my previous concerns. The manuscript is significantly improved and I consider it to be worthy of publication.

---

## [Editor Report · Decision Letter 3]

4 Aug 2021

Dear Dr. Kallapur,

On behalf of my colleagues and the Academic Editor, Gordon Smith, I am pleased to say that we can in principle offer to publish your Research Article "The induction of preterm labor in Rhesus macaques is determined by the strength of immune response to intrauterine infection" in PLOS Biology, provided you address any remaining formatting and reporting issues. These will be detailed in an email that will follow this letter and that you will usually receive within 2-3 business days, during which time no action is required from you. Please note that we will not be able to formally accept your manuscript and schedule it for publication until you have made the required changes.

PRESS

Sincerely, 

Paula 

---

Paula Jauregui, PhD 

Associate Editor 

PLOS Biology
